# Furosemide and the Symptom Burden: The Potential Mediating Role of Uremic Toxins in Patients with CKD

**DOI:** 10.3390/toxins17110541

**Published:** 2025-11-01

**Authors:** Margaux Costes-Albrespic, Sophie Liabeuf, Islam-Amine Larabi, Solène M. Laville, Bénédicte Stengel, Abdou Y. Omorou, Luc Frimat, Jean-Claude Alvarez, Ziad A. Massy, Natalia Alencar de Pinho

**Affiliations:** 1Centre for Research in Epidemiology and Population Health (CESP), INSERM UMRS 1018, Paris-Saclay University, Versailles Saint-Quentin University, F-94807 Villejuif, France; costesalbrespic@hotmail.fr (M.C.-A.); ziad.massy@auraparis.org (Z.A.M.); 2Pharmaco-Epidemiology Unit, Department of Clinical Pharmacology, Amiens-Picardie University Medical Centre, F-80054 Amiens, France; liabeuf.sophie@chu-amiens.fr (S.L.);; 3MP3CV Laboratory, Jules Verne University of Picardie, F-80054 Amiens, France; 4Department of Pharmacology and Toxicology, CARNOT Personalized Medicine Platform, Raymond Poincaré Hospital, Assistance Publique–Hôpitaux de Paris, F-92380 Garches, France; 5Équipe MOODS, Centre for Research in Epidemiology and Population Health (CESP), French National Institute of Health and Medical Research (Inserm) Unit 1018, Paris-Saclay University, Versailles Saint-Quentin University, F-92380 Garches, France; 6UMR 1319 INSPIIRE, French National Institute of Health and Medical Research (Inserm), Université de Lorraine, F-54500 Vandoeuvre-Lès-Nancy, France; 7Clinical Investigation Center in Clinical Epidemiology, French National Institute of Health and Medical Research (Inserm), University Hospital of Nancy (CHRU Nancy), Université de Lorraine, F-54500 Vandoeuvre-Lès-Nancy, France; 8Nephrology Department, University Regional Hospital of Nancy, F-54500 Vandoeuvre-Lès-Nancy, France; 9AURA (Association pour l’Utilisation du Rein Artificiel), F-75010 Paris, France; 10Nephrology Department, Ambroise Paré University Hospital, Assistance Publique–Hôpitaux de Paris, F-92100 Boulogne-Billancourt, France

**Keywords:** chronic kidney disease, furosemide, loop diuretic, uremic toxin, symptom burden

## Abstract

Furosemide appears to contribute to the accumulation of protein-bound uremic toxins (PBUTs) and to induce adverse drug reactions. We investigated the extent to which the association between the furosemide dose and serum PBUT concentrations mediates the relationship between the furosemide dose and the symptom burden in patients with chronic kidney disease (CKD). This cross-sectional analysis included patients with CKD stages 2 to 5 from the CKD-REIN cohort and with data on the baseline serum concentrations of the free fractions of indoxyl sulphate (IS), kynurenine (KYN), p-cresyl sulphate (PCS), and indole-3-acetic acid (IAA), as measured by liquid chromatography–tandem mass spectrometry. The symptom burden was also assessed with a modified (8-item) symptom subscale from the Kidney Disease Quality of Life-36 (e.g., muscle soreness, cramps, itchy skin, dry skin, dizziness, appetite, numbness, and nausea). We used beta regressions to model the association between the furosemide dose and the symptom burden and used structural equation models to quantify the mediating effect of PBUT on this association. Among the 2053 included patients (males: 66%, median age: 68; mean estimated glomerular filtration rate: 35 mL/min/1.73 m^2^), those prescribed high-dose furosemide (>120 mg/day) had higher symptom burden than those not prescribed furosemide (i.e., a 5.67-point lower symptom score, 95%CI 1.41–9.93). The sum of PBUTs explained 3.78% (95%CI 0.10–18.01%) of this association. Similar results were observed for IS, KYN, and IAA, considered separately, but not for PCS, whose estimated mediation effect was nearly null. Although high-dose furosemide was associated with a greater symptom burden in patients with CKD, mediation by PBUT accumulation appeared to be minimal.

## 1. Introduction

Patients with chronic kidney disease (CKD) often report a significant symptom burden, which results from a high prevalence of comorbidities, polypharmacy, and kidney dysfunction itself [1,2,3,4]. The accumulation of uremic toxins (UTs) appears to hasten the development of psychoneurological symptoms (such as pain, fatigue, anxiety, depression, sleep disorders, and cognitive dysfunction) through the compounds’ pro-inflammatory properties [1]. UT accumulation might also contribute to other symptoms in patients with CKD, such as anorexia, nausea, vomiting, and pruritus [1,5,6,7].

The loop diuretic furosemide (commonly prescribed in CKD) is indicated in cases of fluid overload and high blood pressure [8]. However, furosemide use is associated with a range of sometimes symptomatic adverse events (Table 1) [9]. The hypotension caused by diuresis and vasodilatation [9] is known to induce dizziness and/or numbness [10]. Serum electrolyte concentration imbalances and metabolic alkalosis [9] can provoke many symptoms, including nausea, cramps, muscle soreness, lack of appetite, dizziness, numbness, and dry skin [8,11,12,13].

Recently, our group described elevated serum levels of protein-bound UTs (PBUTs) excreted by organic anion transporters 1 and 3 (OAT1/OAT3)—indoxyl sulphate (IS), kynurenine (KYN), p-cresyl sulphate (PCS), and indole-3-acetic acid (IAA)—in patients prescribed furosemide, particularly at doses above 120 mg/day [14]. Although it is well established that CKD leads to elevated levels of UTs, we hypothesised that furosemide further impairs the kidney’s clearance of specific UTs independently of renal function. The resulting additional accumulation might exacerbate the drug’s toxicity and contribute to symptoms.

The main objectives of the present study of CKD patients were therefore to (i) evaluate the association between the furosemide dose and the symptom burden, and (ii) assess the extent to which the latter association is mediated by free-fraction PBUT concentrations (IS, KYN, PCS, IAA, and their sum, referred to hereafter as ∑UTs).

## 2. Results

### 2.1. Characteristics of the Study Participants

Among the 2053 patients included (Appendix A), the median age was 68 years (interquartile range, 61–76), 1358 (66%) were males, 1067 (52%) had a history of cardiovascular disease, and 818 (40%) had diabetes (Table 2). Data on missing information in the furosemide dose groups are presented in Appendix A. Furosemide was prescribed at a daily dose of 10–40 mg, 60–120 mg, or >120 mg to 400 (19%), 165 (8%), and 131 (6%) patients, respectively. Patients prescribed higher doses of furosemide had worse kidney function (with an estimated glomerular filtration rate (eGFR) ranging from 28 mL/min/1.73 m^2^ in patients prescribed a furosemide dose >120 mg/day to 37 mL/min/1.73 m^2^ in those without a furosemide prescription), a higher prevalence of comorbidities (50% and 5% with a history of heart failure (HF), respectively), higher UT concentrations, and a greater symptom burden according to both the 8- and 11-item symptom scores (Table 2).

### 2.2. Symptom Score Items at Baseline

Patients receiving a higher dose of furosemide were more likely to report being very to extremely bothered by the three symptoms considered to be indications for furosemide: shortness of breath, feeling “washed out” or drained, and chest pain. The proportions of these three symptoms increased with the dose of furosemide, from 12.5% to 35.4%, 19.0% to 33.8%, and 1.9% to 5.9%, respectively (Figure 1). Except for cramps, patients prescribed higher doses of furosemide also reported being more bothered by dry skin, soreness in muscles, and numbness in the hands and feet, in particular. The most strongly correlated symptoms were shortness of breath and feeling washed out or drained (Spearman’s correlation coefficient *ρ*: 0.50). Pairwise correlations between symptoms that were considered (or not) to be indications for furosemide were the highest for muscle soreness and feeling washed out or drained (*ρ*: 0.48), numbness in hands and feet with feeling washed out or drained (*ρ*: 0.42), and faintness or dizziness with feeling washed out or drained (*ρ*: 0.41, Appendix A).

### 2.3. Association Between the Furosemide Dose and the Symptom Score

Patients prescribed >120 mg/day of furosemide had significantly lower symptom scores (predicted absolute difference, 5.67; 95%CI 1.41–9.93) in adjusted models (Table 3), compared with those not prescribed furosemide. This association did not appear to differ by sex (Appendix A) but was slightly attenuated after additional adjustment for ∑UTs (predicted absolute difference, 5.48, 95%CI 1.22–9.74). These results were consistent with those obtained when the furosemide dose was modelled as a continuous variable with a smoothing function (Appendix A). Further adjustments for educational level, adherence, systolic blood pressure, serum calcium, serum potassium and the haemoglobin concentration did not alter the association (Appendix A).

The extent to which patients were bothered by dry skin and numbness in hands and feet had a graded association with the furosemide dose, with odds ratios (ORs) (95%CI) for an ordered increase in dose category of 1.88 (1.30 to 2.73) and 2.19 (1.51 to 3.17), respectively (Appendix A).

### 2.4. The Mediating Effect of the Furosemide Dose on the Symptom Burden, via an Accumulation of Uremic Toxins

∑UTs partially mediated furosemide’s effect on the symptom burden (according to the 8-item symptom score) by 14.78% (95%CI, 7.58%; 27.92%) in a crude analysis (Appendix A) and 3.60% (95% IC, 0.27%; 21.39%) in adjusted analyses (Figure 2). Patients prescribed >120 mg/day of furosemide had significantly higher concentrations of ∑UTs (+13.46%, 95%CI 2.03%; 27.23%), compared with those prescribed lower doses or none. A two-fold increment in ∑UTs was associated with a decrement of −0.96 (95%CI −1.96; −0.07) in the symptom score. Similar proportions of mediation were observed for IS (5.33%, 95% IC 0.92% to 20.29%), IAA (3.52%, 0.25% to 19.96%), and KYN (3.06%, −0.07% to 13.67%), but not for PCS (−0.03%, −3.18% to 3.07%) (Appendix A). These results were robust when the functional form of the furosemide dose was specified in different ways (Appendix A).

## 3. Discussion

This cross-sectional study of a large cohort of patients with CKD stages G2 to G5 highlighted the strong association between the furosemide dose (and especially high doses, i.e., >120 mg/day) and the symptom burden, independently of CKD severity, comorbidities, and the overall prescription burden (the number of prescription medications). This association was slightly but significantly mediated by the association between the furosemide dose on one hand and ∑UTs and the free-fraction concentrations of IS, IAA, and KYN (but not PCS) on the other. These results are important for understanding possible mechanisms of iatrogenesis in CKD in general and those involving UT accumulation in particular.

The association between the furosemide dose and the symptom burden was expected because furosemide is known to cause hypotension and metabolic imbalances, which can manifest themselves through various symptoms [8,9,10,11,12,13]. Although the overall symptom score was lower at higher furosemide doses, analysis of individual symptom score items showed that after adjusting for confounders and multiple comparisons, patients prescribed higher furosemide doses specifically reported being more bothered by dry skin and numbness in the hands and feet. In furosemide users, dry skin has conventionally been attributed to dehydration, while numbness in the extremities has been attributed to hypotension, hypocalcaemia, and/or hypokalaemia [8,9,10,11,12,13]. However, further adjustment for these variables did not alter the observed association (Appendix A). This might mean that within-subject variability is greater than between-subject variability.

The interactions between UTs and drugs are complex and are not limited to the elimination phase. Indeed, the results of a recent study of anuric patients undergoing haemodialysis suggested that UTs interact with loop diuretics at the distribution stage [15]; this is in line with our previous report [14] in which higher furosemide doses were associated with elevated concentrations of free ∑UTs, IS, KYN, PCS, and IAA. Building on these findings and the known relationship between UT concentrations and the symptom burden [16], we determined the extent to which the relationship between the furosemide dose and the symptom burden was explained by the association between the furosemide dose and elevated PBUT concentrations. We found that the relation between the furosemide dose and PBUT accumulation accounted for only 3.78% of the relation between the furosemide dose and the symptom burden. Furthermore, our results suggest that the PBUTs studied here have little clinical relevance in the context of furosemide-related symptoms. However, we cannot rule out significant mediation by PBUTs as a whole, given that (i) we were able to assay only five of the hundreds of known PBUTs and (ii) other PBUTs are perhaps yet to be discovered.

In the present study, PCS did not seem to have a mediating effect on the relationship between the prescribed furosemide dose and symptom burden. This was probably because PCS concentrations were not significantly associated with the symptom burden. This finding may reflect differences in metabolic origin: PCS is derived from the bacterial metabolism of tyrosine, whereas IS, IAA, and KYN are tryptophan-derived UTs [17]. There is also a gap in the literature regarding direct associations between PCS levels and specific symptoms. Only one observational study of elderly adults with CKD identified total PCS as being positively associated with constipation [18]. In contrast, tryptophan-derived UTs have been implicated more directly in neurological and inflammatory pathways, such as activation of the aryl hydrocarbon receptor, endothelial dysfunction, and leukocyte activation—all of which may contribute to symptoms like fatigue, pruritus, and neurological disorders [19]. Other studies showed that the total KYN concentration is associated with gastrointestinal disorders [20]. Both elevated and low total KYN concentrations (relative to those in healthy controls) have been observed in patients suffering from chronic fatigue, but potential confounding bias means that these findings should be interpreted with caution [20]. Besides an electrolyte imbalance, the long-term accumulation of PBUTs (and notably total KYN and free IS) might contribute to oxidative stress and endothelial dysfunction, which can lead to nerve damage and symptoms like numbness and chronic pain [21,22].

Polypharmacy is common in patients with CKD [23]. When reviewing or initiating drugs for patients with polypharmacy, it is therefore important to consider whether or not furosemide might increase symptom burden (via UTs or otherwise). Furosemide is often prescribed for the management of blood pressure and fluid overload in CKD—especially when the eGFR falls below 30 mL/min/1.73 m^2^ [24]. The drug can relieve symptoms such as dyspnoea, chest pain, and fatigue [25,26] but can also induce adverse reactions. Recent evidence suggests that furosemide competes with PBUTs for plasma protein binding sites, which results in a higher free fraction of PBUTs [15,27]. This furosemide property may be therapeutically beneficial via enhanced dialytic clearance of these toxins.

In the present study, we described the extent to which furosemide use was associated with the symptom burden. Other loop diuretics (such as bumetanide or torsemide) might be better pharmacological options than furosemide for managing symptoms, given their higher bioavailability and (for torsemide) longer half-life [28]. However, retrospective, comparative studies have been inconclusive with regard to which specific loop diuretic gives the best long-term clinical outcomes and the best symptom management—perhaps due to confounding by indication [26,29,30,31,32,33,34,35]. For instance, the TRANSFORM-HF study, of 2859 patients discharged from hospital after admission for HF, did not show the superiority of torsemide over furosemide in terms of symptom scores or quality of life over 12 months [26]. Prospective studies are needed to determine whether one specific loop diuretic offers greater symptom control than others in patients with CKD. Given that sodium-glucose cotransporter 2 (SGLT-2) inhibitors exert beneficial effects on body fluid compartment distribution and volume status (mainly with regard to extracellular water, body fat mass, and visceral fat) without the loss of skeletal muscle mass [36], it would be of interest to determine whether combining an SGLT-2 inhibitor with low-dose furosemide in patients with CKD and fluid overload is an effective, safe way of achieving euvolemia. Furthermore, SGLT-2 inhibitor use by patients with CKD is associated with a distinct composition of the gut microbiota and lower concentrations of uremic solutes [37].

Our study had several strengths. It was based on a large cohort of patients with a confirmed diagnosis of CKD and recruited from a nationally representative sample of nephrology outpatient facilities. Extensive data collection enabled us to identify the furosemide dose prescribed and account for potential confounders, such as the eGFR and comorbidities. UT concentrations were measured in the same central laboratory, using a robust, validated, ultra-high-performance liquid chromatography tandem mass spectrometry (LC-MS/MS) technique. Lastly, our outcomes were based on the symptom burden subscale of the KDQOL; although not the most sensitive or specific, the KDQOL is the instrument most widely used for health-related quality of life assessment in CKD and enables between-study comparisons [38].

Our study also had some limitations. Firstly, the study’s cross-sectional design prevented us from assessing the chronological nature of the relationship between the furosemide dose, UTs, and symptoms. Secondly, the items that are typically indications for furosemide were excluded from the symptom score. These items were only weakly correlated with the symptoms studied here (Spearman’s *ρ* < 0.5), which suggests that the observed association was not due to indication bias. Although we adjusted our models for several factors that may influence furosemide prescriptions, UT levels and symptom score, we cannot rule out residual confounding—particularly from markers of tubular excretion. Lastly, our definition of furosemide exposure was based on prescriptions and not actual medication intake. Data on blood or urine levels of furosemide (i.e., possible markers of adherence) were not available. However, given that furosemide is a prescription-only drug in France, the misclassification of exposure (if any) was more likely to result in underestimation of the strength of the relationship between the furosemide dose, UT concentrations, and symptoms.

In conclusion, the prescribed furosemide dose was associated in a graded manner with the symptom burden. Only a very small proportion of this association appeared to be mediated by the accumulation of specific UTs, including IAA, IS, and KYN. Therefore, furosemide-related symptoms are unlikely to be explained by the accumulation of the free fraction of the studied UTs. Given the high symptom burden associated with the prescription of high doses of furosemide and uncertainty over furosemide’s risk-benefit ratio, alternative drugs should be explored in future studies.

## 4. Materials and Methods

### 4.1. Data Source and Population

The CKD–Renal Epidemiology and Information Network (CKD-REIN) is a prospective cohort study conducted in a sample of 40 nephrology practices in France that were nationally representative in terms of geographic distribution and legal status. From 2013 to 2016, we included 3033 patients with CKD stages 2 to 5 who were not on maintenance dialysis and had not undergone kidney transplantation. The CKD-REIN study’s rationale, design, and methods have been described in detail elsewhere [27]. The protocol was approved by the French National Institute of Health and Medical Research’s independent ethics committee (CEE IRB00003888 Paris, France, on 13 June 2013), and the study was registered at ClinicalTrials.gov (NCT03381950). All the study participants were aged 18 or over and gave their written, informed consent.

In the present analysis, we included patients who had complete information on PBUT levels (n = 2406) and symptom score (n = 2575), each collected within 3 months of baseline. Patients with missing data on the prescribed dose of furosemide were excluded (n = 15). A total of 2053 patients were included in the final analysis (Appendix A).

### 4.2. Furosemide Prescription

At the inclusion visit, participants were asked to bring all their drug prescriptions (issued by any physician) over the past 3 months. Drugs were then coded by clinical research associates with an electronic case report form linked to the international Anatomical Therapeutic and Chemical (ATC) thesaurus [39]. For each drug prescription, the trade name, international non-proprietary name, ATC class, unit dose, prescribed daily dose, pharmaceutical formulation, and administration route were available. Furosemide was identified by the C03CA01 ATC code. In accordance with our previous study, the corresponding dose was categorised into four categories: None, 10–40 mg, 60–120 mg, >120 mg per day; there was no furosemide prescription at doses below 10 mg or between 40 and 60 mg. This categorisation was determined based on clinical practice and the quantile distribution of the variable.

### 4.3. Symptom Score

All study participants were asked to complete self-questionnaires, including the Kidney Disease Quality of Life-36 (KDQOL-36) [40], which has been validated for the assessment of health-related quality of life in patients with CKD. The KDQOL-36 is divided in five subscales: the burden, effects and symptoms of kidney disease, and the Short Form (SF)-12 mental and physical component summaries. Here, we focused on the symptom subscale based on 11 items: muscle pain or soreness, chest pain, cramps, itchy skin, dry skin, shortness of breath, faintness or dizziness, lack of appetite, feeling washed out or drained, numbness in hands or feet, and nausea or upset stomach. Patients were asked: “During the past four weeks, to what extent were you bothered by (the particular symptom)?” Responses were recorded using a five-modality Likert scale: “not at all”, “somewhat”, “moderately”, “very much”, or “extremely bothered”. Each symptom represents one item of the score. In accordance with Hays et al.’s KDQOL-36 scoring manual [40], the symptom score was calculated as follows: if fewer than 50% of the items were missing, the score was computed as the mean of non-missing items multiplied by the total number of items. The symptom score ranged from 0 (worst possible) to 100 (best possible). In our study, this score is referred to as the “11-item symptom score”. To limit the main analysis to symptoms potentially arising from furosemide-associated adverse events (Table 1, as opposed to symptoms which are indications for furosemide prescription [25,26]), we excluded three symptom items (chest pain, shortness of breath, and feeling washed out or drained) and considered them to be missing data. “Feeling washed out or drained” corresponds to fatigue in the Kansas City Cardiomyopathy Questionnaire, a patient-reported measure of symptom burden and quality of life for patients with HF; we included it as a potential furosemide indication in order to better reflect clinical reasoning and avoid missing relevant cases. This symptom score is denoted hereafter as the “8-item symptom score”. As with the 11-item symptom score, lower scores indicate a greater symptom burden.

### 4.4. Serum Concentrations of UTs and Other Centralised Measurements

At baseline, serum samples were collected in the morning, while fasting, immediately stored at 4 °C, and aliquoted within 6 h without additional processing. The samples were stored at −80 °C in a biological resource centre (Biobanque de Picardie, BRIF number: BB-0033-00017, Amiens, France) and shipped frozen to Paris (France) for analysis. Serum concentration of high-sensitivity C-reactive protein was assayed by a chemistry analyser (Architect C16000, Abbott, Chicago, IL, USA), and serum albumin concentrations by immunoturbidimetry (Atellica^®^ CH, Siemens, Erlangen, Germany). Centralised isotope dilution mass spectrometry-traceable creatinine concentration was determined with enzyme assays, and eGFR was estimated with the 2009-CKD-Epidemiology Collaboration equation. PBUT fractions in serum were assayed through a validated ultrahigh-performance liquid chromatography tandem mass spectrometry (LC-MS/MS) technique, as described previously [41]. Free serum concentrations of IS, KYN, PCS, IAA and their sum were selected for this study based on findings of a previous study of ours suggesting that furosemide doses above 120 mg/day are associated with their accumulation [14]. Notably, no association was found between furosemide dose and total UT concentrations. Therefore, we focused exclusively on free UT concentrations.

### 4.5. Covariates

Baseline data (including sociodemographic characteristics, medical histories, and laboratory data) were collected from interviews, medical records, and self-questionnaires by clinical research associates. Sex was defined as sex assigned at birth. Diabetes was defined as the prescription of a glucose-lowering drug, a glycated haemoglobin concentration ≥6.5%, a fasting glucose concentration ≥7 mmol/L, or a non-fasting glucose concentration ≥ 11 mmol/L. The history of cardiovascular disease was assessed through medical records and included heart failure (HF), coronary artery disease (CAD), cerebrovascular disease, peripheral arterial disease (PAD), and cardiac rhythm disorders. Urine albumin-to-creatinine ratio (uACR) was measured or estimated from the protein-to-creatinine ratio [42]; both were performed in patients’ usual hospital central laboratories and/or private medical laboratories, as was haemoglobin. Height and weight data, measured by nephrologists or outpatient nurses during the enrolment visit, were used to calculate the body mass index (BMI, kg/m^2^). Obesity was defined by a BMI ≥ 30 kg/m^2^.

### 4.6. Statistical Analyses

Continuous variables were reported as the median [interquartile range] or the mean (standard deviation) as appropriate (according to their skewness), and categorical variables were reported as the frequency (percentage). The patients’ characteristics were described by furosemide dose category. Bar plots were used to depict symptom severity by furosemide dose category. Pairwise correlations between items of the symptom score were assessed by calculating Spearman’s rank correlation coefficient *ρ*. Data on PBUT concentrations and other variables with a skewed distribution were log-transformed.

To evaluate whether a PBUT mediation effect was plausible, we assessed the association between the furosemide dose category and the 8-item symptom score before and after adjustment for ∑PBUTs (Baron & Kenny’s method) [43]. Given the symptom score’s left-skewed distribution, beta regression models with a logit link were used. Since the beta regression requires values between 0 and 1, the 8-item symptom score was transformed to fit within this range (Supplemental Methods) [44]. The two models were adjusted for clinically relevant factors identified in the literature, including age, sex, obesity, uACR, eGFR, smoking status, haemoglobin concentration, diabetes, a history of HF, CAD, cerebrovascular disease, PAD, and the number of co-prescribed medications (Appendix A). We examined whether this association differed according to sex by including a multiplicative interaction term between sex and the furosemide dose. Ordinal logistic regressions were used to assess the association between the furosemide dose and each of the 11 items in the symptom score.

We then implemented structural equation modelling (SEM, using the lavaan package in R [45]) to assess total, direct and indirect effects of furosemide on the 8-item symptom score mediated by free concentrations of IS, KYN, PCS, IAA or their sum (∑PBUTs). This approach supports binary and continuous independent variables without the need fpr a smoothing function, we thus categorised furosemide doses into two groups (≤120 mg/day and >120 mg/day) in a main analysis [45]. Two regression models were specified: firstly, a model for the 8-item symptom score adjusted for the log-PBUT concentration and the covariates used in the beta regression; secondly, a model for the log-PBUT concentration, adjusted for age, sex, obesity, uACR, eGFR, smoking status, diabetes, history of HF, CAD, cerebrovascular disease, PAD, the number of co-prescribed medications, log-CRP, and serum albumin, in accordance with our previous study [14]. Sensitivity analyses included modelling the association between the symptom score and the furosemide dose as a continuous variable with a smoothing function in beta regression models (natural splines, with knots at 40, 80, 100 and 120 mg), as a linear variable (increments of 20 mg) and an ordinal variable (for an increase in one dose category out of four) in SEM. Furthermore, beta regression models were further adjusted for adherence, educational level, systolic blood pressure, and serum calcium and potassium concentrations.

Assuming that data were missing at random, we performed multiple imputation with chained equations (MICE, 30 generated datasets) with the MICE package in R software (version 4.1.2) [46,47] for beta and ordinal logistic regression models and the full information maximum likelihood method for SEM. For more details of the statistical analyses, please refer to the Supplemental Methods. The threshold for statistical significance was set to *p* < 0.05. All statistical analyses were performed with R software (version 4.1.2).

## Figures and Tables

**Figure 1 toxins-17-00541-f001:**
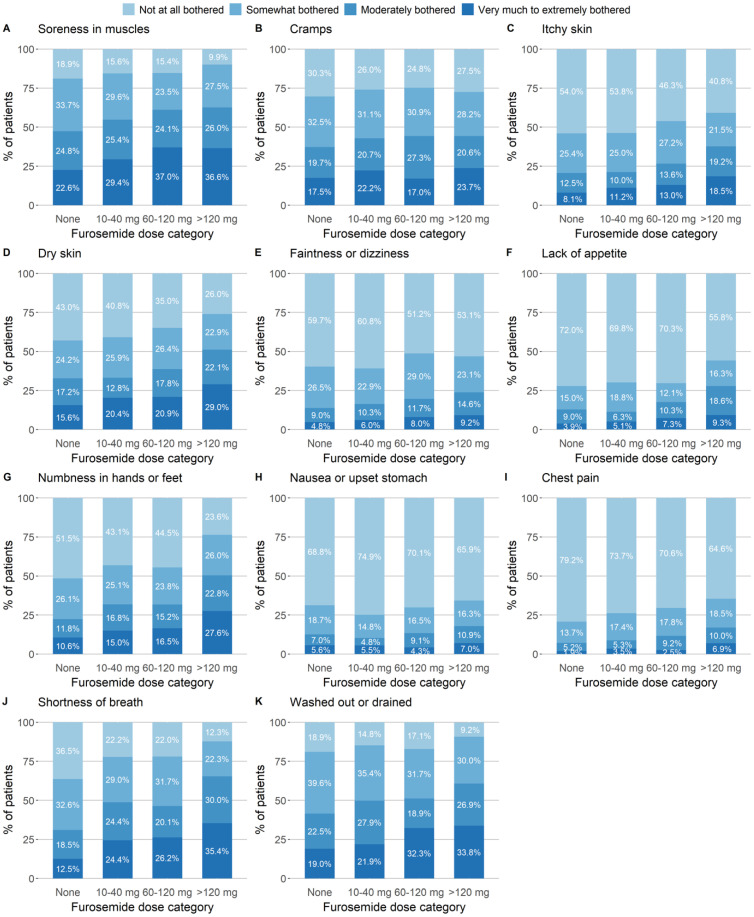
Symptom severity at baseline, by furosemide dose category and based on the symptom subscale of the KDQOL-36. Patients were asked: “During the past four weeks, to what extent were you bothered by [the particular symptom]?” Answers were given on a five-modality Likert scale: “not at all”, “somewhat”, “moderately”, “very much”, or “extremely bothered”. The categories “very much” and “extremely” were pooled into a single category.

**Figure 2 toxins-17-00541-f002:**
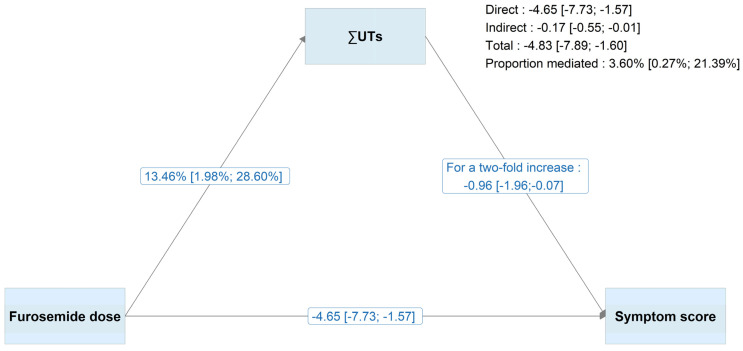
Results of the adjusted structural equation modelling analyses for ∑UTs and with the furosemide dose as a binary variable (≤120 mg/day vs. >120 mg/day). Maximum likelihood estimation with bootstrapping (1000 replications) was applied. The symptom score was based on eight selected symptoms (muscle soreness, cramps, itchy skin, dry skin, faintness or dizziness, lack of appetite, numbness in hands or feet, and nausea or upset stomach), and ∑UTs were log-transformed. For interpretability, estimates were transformed by using the exponential function or multiplying by log(2), in line with standard conventions for log-linear models. The model of the 8-item symptom score was adjusted for age, sex, obesity, log-uACR, eGFR, smoking status, haemoglobin concentration, diabetes, history of HF, coronary heart disease, cerebrovascular disease, peripheral artery disease, and the number of prescribed medications. The model of the ∑UT concentration was adjusted for the same covariates (apart from haemoglobin), log-CRP, and serum albumin. Abbreviations: ∑UTs, sum of the free concentrations of indoxyl sulphate, kynurenine, p-cresyl sulphate and indole-3-acetic acid.

**Table 1 toxins-17-00541-t001:** Effects of furosemide on serum electrolytes, acid-base balance, and other factors, and the associated symptoms.

Potential Symptoms Furosemide Effect	Soreness in Muscle	Cramps	Dry Skin	Itchy Skin *	Nausea or Upset Stomach	Lack of Appetite	Faintness or Dizziness	Numbness in Hands or Feet
Hypotension							x	x
Metabolic Alkalosis	x	x	x		x	x	x	x
Dehydration (including hypernatremia)		x	x		x	x	x	
Hypokalaemia	x	x						x
Hypocalcaemia		x						x
Hypomagnesemia		x			x	x		
Hyperuricemia	x				x	x	x	

This table is based on references [5,8,9,10,11,12,13]. * No studies report ‘itchy skin’ as a common symptom of furosemide.

**Table 2 toxins-17-00541-t002:** Patient’s characteristics at baseline, overall and by furosemide dose category.

Characteristic	Overall N = 2053	Furosemide Prescription				Missing Data (N, %)
		None N = 1357	10–40 mg/day N = 400	60–120 mg/day N = 165	>120 mg/day N = 131	
Age (years), median [IQR]	68 [61; 76]	67 [58; 74]	71 [65; 79]	72 [66; 80]	71 [67; 77]	0, 0%
Men	1358 (66%)	886 (65%)	264 (66%)	114 (69%)	94 (72%)	0, 0%
Smoker	232 (11%)	164 (12%)	39 (10%)	17 (10%)	12 (9%)	11, 0.5%
eGFR (mL/min/1.73 m^2^), mean (SD)	35 (13)	37 (13)	31 (12)	29 (11)	28 (11)	11, 0.5%
Albumin-to-creatinine ratio (mg/g), median [IQR]	104 [20; 496]	81 [17; 455]	133 [24; 531]	207 [34; 768]	230 [42; 778]	305, 15%
Diabetes	818 (40%)	423 (31%)	197 (49%)	102 (62%)	96 (73%)	5, 0.2%
Cardiovascular history	1067 (52%)	559 (41%)	272 (68%)	124 (76%)	112 (85%)	12, 0.6%
Coronary artery disease	489 (24%)	225 (17%)	129 (33%)	73 (45%)	62 (48%)	16, 0.8%
Heart failure	261 (13%)	73 (5%)	76 (19%)	46 (28%)	66 (50%)	3, 0.1%
Cerebrovascular disease	220 (11%)	122 (9%)	47 (12%)	28 (17%)	23 (18%)	17, 0.8%
Peripheral artery disease	320 (16%)	151 (11%)	91 (23%)	43 (26%)	35 (27%)	15, 0.7%
Obesity (BMI ≥ 30 kg/m^2^)	711 (35%)	362 (27%)	175 (45%)	85 (52%)	89 (68%)	35, 1.7%
C-reactive protein (mg/L), median [IQR]	2.3 [1.1; 5.0]	2.0 [0.9; 4.2]	2.6 [1.3; 5.4]	2.8 [1.5; 6.7]	5.1 [2.0; 9.1]	105, 5.1%
Serum albumin (g/L), mean (SD)	41.0 [38.5; 43.3]	41.3 [39.0; 43.6]	40.6 [37.9; 42.7]	40.0 [37.5; 42.5]	39.0 [35.6; 41.5]	7, 0.3%
Haemoglobin (g/dL), mean (SD)	13.09 (1.66)	13.29 (1.62)	12.78 (1.79)	12.90 (1.47)	12.35 (1.57)	30, 1.5%
Serum potassium (mmol/L)	4.53 (0.51)	4.54 (0.50)	4.61 (0.54)	4.45 (0.51)	4.29 (0.51)	6, 0.3%
Serum calcium (mmol/L)	2.35 (0.13)	2.36 (0.12)	2.35 (0.14)	2.34 (0.14)	2.29 (0.18)	66, 3.2%
Number of drugs prescribed, median [IQR]	8 [5; 10]	6 [4; 9]	9 [8; 12]	10 [8; 13]	12 [9; 14]	8, 0.4%
Uremic toxins, median [IQR]						
Indoxyl sulphate (µM)	0.24 [0.13; 0.44]	0.21 [0.12; 0.38]	0.28 [0.16; 0.50]	0.33 [0.17; 0.63]	0.45 [0.23; 0.80]	0, 0%
Kynurenine (µM)	0.61 [0.43; 0.86]	0.56 [0.40; 0.79]	0.71 [0.51; 0.94]	0.75 [0.55; 0.96]	0.85 [0.59; 1.22]	0, 0%
P-cresyl sulphate (µM)	0.99 [0.45; 1.90]	0.82 [0.37; 1.57]	1.33 [0.65; 2.19]	1.36 [0.86; 2.93]	1.65 [0.80; 3.46]	0, 0%
Indole-3-Acetic Acid (µM)	0.18 [0.13; 0.29]	0.18 [0.13; 0.27]	0.18 [0.14; 0.29]	0.22 [0.14; 0.33]	0.26 [0.17; 0.35]	0, 0%
∑UTs (µM)	2.15 [1.37; 3.51]	1.85 [1.22; 3.02]	2.57 [1.68; 4.23]	2.87 [1.97; 4.67]	3.44 [2.28; 5.30]	0, 0%
8-item symptom score, median [IQR]	78 [69; 88]	81 [69; 91]	78 [65; 88]	75 [66; 84]	72 [59; 81]	0, 0%
11-item symptom score, median [IQR]	80 [66; 89]	80 [68; 89]	77 [64; 86]	73 [61; 84]	68 [58; 80]	0, 0%

Abbreviations: BMI, body mass index; eGFR, estimated glomerular filtration rate; IQR, interquartile range; SD, standard deviation; ∑UTs, sum of the free concentration of indoxyl sulphate, kynurenine, p-cresyl sulphate and indole-3-acetic acid.

**Table 3 toxins-17-00541-t003:** The association between the symptom score and the furosemide dose category, adjusted for potential confounders and for ∑UTs.

	Furosemide Dose Category	Furosemide Dose Category + ∑PBUTs
	Exp (Estimate) (95%CI), Reference: No Furosemide Prescription	
Crude		
10–40 mg/day	0.90 (0.81–0.99)	0.94 (0.85–1.04)
60–120 mg/day	0.80 (0.69–0.92)	0.84 (0.73–0.97)
>120 mg/day	0.57 (0.49–0.67)	0.62 (0.53–0.72)
Adjusted *		
10–40 mg/day	1.04 (0.94–1.16)	1.05 (0.94–1.16)
60–120 mg/day	0.94 (0.81–1.09)	0.95 (0.82–1.10)
>120 mg/day	0.73 (0.62–0.87)	0.74 (0.62–0.88)
	Predicted absolute difference (95%CI)	
Adjusted *		
None—10–40 mg/day **	−0.69 (−2.92–1.54)	−0.76 (−2.99–1.47)
None—60–120 mg/day **	1.01 (−2.34–4.35)	0.91 (−2.43–4.24)
None—>120 mg/day **	5.67 (1.41–9.93)	5.48 (1.22–9.74)

Exponentiated estimates from beta regression with a logit link are interpretable as the ratio (mean symptom score)/(1 − mean symptom score). Values < 1 indicate a lower symptom score (i.e., a higher symptom burden), while values > 1 indicate a higher symptom score (i.e., lower symptom burden). ∑UTs were log transformed. * Adjusted for age (natural splines, with knots at 50, 70 and 80 years), sex, obesity, log urinary albumin-to-creatinine ratio (uACR), eGFR, smoking status, diabetes, history of HF, coronary heart disease, cerebrovascular disease, peripheral disease, and the number of co-prescribed medications. ** Values > 0 indicate a lower symptom score (i.e., a higher symptom burden), while values < 0 indicate a higher symptom score (i.e., a lower symptom burden). Abbreviations: CI, confidence interval; eGFR, estimated glomerular filtration rate; uACR, urinary albumin-to-creatinine ratio; ∑PBUTs, sum of the free concentrations of IS, KYN, PCS and IAA.

## Data Availability

Processing of the data supporting the study findings are under the responsibility of the Institut National de la Santé et de la Recherche Medicale (Inserm), France and complies with the European Regulation (EU) 2016/679 (General Data Protection Regulation) related to “the protection of natural persons with regard to the processing of personal data and rules relating to the free movement of personal data”. In compliance with Inserm standard data sharing processes and the agreements obtained from the “Commission nationale de l’informatique et des libertés (CNIL_DR-2012-469)” and the ethics committee (IRB00003888 and CCTIRS12.360/CPP), the data can be made available upon request to the study coordinating centre Via ckdrein@inserm.fr. The code used in the analyses is stored within servers at the Centre de recherche en Epidémiologie et Santé des Population (CESP, Univ Paris-Saclay, Inserm, Villejuif, France) and can also be made available upon request. Any relevant summary statistics for the article are already included within the main article and will be publicly available once the article is published.

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
