# Peer review of "Furosemide and the Symptom Burden: The Potential Mediating Role of Uremic Toxins in Patients with CKD"

_toxins, 2025, doi:10.3390/toxins17110541_

Round 1
Reviewer 1 Report
Comments and Suggestions for Authors
The authors present a cross-sectional secondary analysis of CKD-REIN assessing the association of high-dose furosemide use with symptom burden among people with CKD stages 2-5. They report that furosemide >120 mg was associated with higher symptom burden, but that this was only slightly mediated by higher levels of uremic toxins.
Strengths: The concept of the study is interesting. The data are well presented and extensive. The authors present missingness for each variable in Table 2.
Opportunities for improvement:
- In the introduction there is a parenthetical right before reference 9 that appears blank and may be typo. There is another similar one in section 4.3.
- In Table 1, each row under “metabolic alkalosis” is indented. It is not clear to me why this would be the case, as the listed electrolyte abnormalities do not necessarily cooccur with metabolic alkalosis. Please clarify.
- Figure S1 is confusing. Please be more clear about who was included in each box that currently only list an N value. It is unclear why the exclusion criteria would be applied in parallel and then merged rather than applying them in series.
- There are instances of inappropriately causal language (e.g., “a higher proportion of patients reported being very to extremely bothered by the symptom, as the dose of furosemide increased”). This example implies that individuals were followed over time with escalating doses; a more appropriate way to state this is that people receiving a higher dose of furosemide were more likely to report being very to extremely bothered by the symptom. The language throughout needs to be revised with careful attention to avoiding implications of causality.
- Furosemide dose was divided into categories. Furosemide is frequently dosed more than once per day. Are the dose categories the total daily dose or the single dose amount? In other words, if someone were prescribed 80 mg twice daily, would they be categorized based on 80 mg (single dose amount) or 160 mg (total daily dose)?
- There has been other interesting recent work investigating the effect of loop diuretic on protein binding (and thus clearance) of uremic toxins in anuric people on hemodialysis (PMID 40671697). In people with non-dialysis CKD whose clearance is determined by their eGFR more so than the quantity of unbound drug, use of high dose loop diuretic may competitively inhibit uremic binding sites for uremic toxins and account for some of the association of higher dose of diuretic with higher free serum concentrations of uremic toxins. It is, therefore, interesting and novel that the authors demonstrate that increased proportion of free (unbound) uremic toxins only slightly mediate the association of diuretic use with symptom burden (which builds on prior studies showing that total PBUT levels are associated with increased uremic symptom burden, e.g., PMID 38266973). The manuscript misses an opportunity to discuss this as a possible mechanism explaining their findings.
- Related to point 6, in the discussion the authors should specify whether prior studies measured total or free levels of PBUT to properly contextualize their findings.
- The authors conducted mediation analysis in cases where the underlying assumptions were not met (e.g., PCS was not associated with symptom burden). Consider presenting mediation analysis only for variables that meet the underlying assumptions to focus on the most pertinent results.
- The discussion about the effect of furosemide on OAT1/OAT3 and other transporters is interesting. However, the authors point out in the methods that there was no association between furosemide and total UT concentrations (section 4.4). I would argue that this effect on clearance of total UTs is less likely to explain the findings, and favor reasoning related to protein binding (which would more cleanly explain why furosemide dose was not associated with total UT concentration but was associated with free UT concentration and is supported by other data as referenced above).
Reviewer 2 Report
Comments and Suggestions for Authors
This cross-sectional study investigates whether protein-bound uremic toxins mediate the relationship between furosemide dose and symptom burden in chronic kidney disease patients using data from the CKD-REIN cohort. While the research question is clinically relevant and the dataset substantial, several methodological concerns limit the validity and interpretability of the findings.
Major issues:
- Statistical model selection: The choice of beta regression appears unnecessarily complex and counterintuitive given that it required extensive data transformations to meet model assumptions. Linear regression on the raw symptom score (sum of points) or proportional odds logistic regression for ordinal symptom responses would be more straightforward and interpretable alternatives that the authors fail to justify avoiding. Moreover, the complex transformations make the results difficult to interpret clinically - for example, when the authors report "patients prescribed >120 mg of furosemide had significantly lower symptom scores (predicted absolute difference, 5.67 95% CI 1.41-9.93)", readers cannot easily understand what this 5.67-point difference means or assess whether this represents a clinically meaningful change in patient experience.
- Uremic toxin summation methodology: The calculation of ΣUTs by summing mass concentrations (ng/mL) is biochemically inappropriate. Different uremic toxins have vastly different molecular weights, making mass-based summation meaningless for biological interpretation. These should be converted to molar concentrations before summation to provide interpretable results.
- Confounding variable classification: The inclusion of hemoglobin as a confounder is questionable, as it likely represents a mediator rather than a confounder in the causal pathway. Higher furosemide doses cause dehydration, which concentrates hemoglobin levels. The authors would benefit from constructing a directed acyclic graph (DAG) to properly identify confounders versus mediators in their analysis.
- Symptom categorization rationale: The exclusion of "feeling washed out or drained" as a furosemide indication lacks adequate justification. While dyspnea and chest pain clearly relate to fluid overload, the clinical basis for considering fatigue as a primary furosemide indication is unclear and requires explanation.
Minor issues:
- Editorial errors: Lines 36-37 and 286-287 contain incomplete parentheses with missing content and awkward line breaks. Line 40 has an unnecessary period after a reference.
- Table inconsistencies: Table 1 fails to assign any furosemide effect to "itchy skin," creating an incomplete framework. The authors should either identify relevant mechanisms or explicitly acknowledge the absence of established associations.
- Missing data reporting: Table 2 reports missing data percentages only overall rather than stratified by furosemide dose groups, which would be more informative for assessing potential bias patterns.
- While the authors mention using free rather than total uremic toxin concentrations, this important methodological decision deserves greater emphasis throughout the manuscript.
- The manuscript completely overlooks important clinical applications of furosemide's interaction with protein-bound uremic toxins. In dialysis patients, the displacement of PBUTs from albumin binding by furosemide, torsemide, or ibuprofen is considered therapeutically beneficial because it enhances dialytic clearance of these toxins (e.g., doi: 10.1093/ckj/sfaf195). This established clinical practice represents a significant gap in the authors' discussion and suggests that the relationship between loop diuretics and uremic toxins may be more nuanced than presented. The potential benefits of toxin displacement in patients approaching dialysis deserve consideration alongside the symptom burden concerns identified in this study.
Reviewer 3 Report
Comments and Suggestions for Authors
This is a well-designed and rigorously analyzed cross-sectional study investigating whether diuretic furosemide worsen symptoms in CKD patients by increasing the accumulation of specific protein-bound uremic toxins (PBUTs). The key finding is that high-dose furosemide (>120 mg/day) is strongly associated with a greater symptom burden, but this association is only minimally mediated by the accumulation of PBUTs (specifically Indoxyl Sulphate, Kynurenine, and Indole-3-Acetic Acid). Nevertheless, there are still some unresolved issues that require the author to provide explanations. I suggest that it could not be accepted to publish before a major’s revision.
- Did furosemide cause higher toxins and symptoms, or are sicker patients with more symptoms and higher toxin levels simply prescribed more furosemide? The authors provide a strong biological rationale for their proposed direction, but longitudinal data is needed for confirmation.
- Despite adjusting for a comprehensive set of covariates, unmeasured factors (e.g., severity of heart failure beyond a simple diagnosis, dietary habits, exact fluid status) could still influence both furosemide prescription and symptom burden.
- The most important influencing factor of high concentration PBUTs in the blood is the ability of kidneys to excrete toxins after functional damage. Therefore, it is necessary to clarify the kidney's ability to clear these substances and incorporate them into a multi factor analysis system for further analysis. Only by excluding confounding factors as much as possible can the true function of furosemide be clarified.
Reviewer 4 Report
Comments and Suggestions for Authors
I would like to sincerely thank the authors for submitting this important manuscript entitled “Furosemide and Symptom Burden, the Mediating Role of Uremic Toxins in Patients with CKD.” The study addresses a clinically relevant issue: the relationship between loop diuretic use, uremic toxin accumulation, and symptom burden in patients with chronic kidney disease (CKD). Based on a large, well-characterized cohort (CKD-REIN), the work provides valuable insights into iatrogenic mechanisms potentially linked to furosemide therapy.
Strengths of the study include the large sample size (>2,000 patients), centralized and standardized measurement of uremic toxins using LC-MS/MS, the use of a validated symptom scale (KDQOL-36), and robust statistical methods (beta regression and structural equation modeling). The discussion is thorough and connects findings with recent literature and possible therapeutic implications. Nevertheless, some issues need to be addressed before the manuscript can be considered for publication. The cross-sectional design is acknowledged, but the discussion could better emphasize causal limitations and the need for longitudinal or interventional studies.
Comments
Comment 1 (Introduction, lines 29–33):
The statement that uremic toxin accumulation “seems to play a role” in psycho-neurological symptoms lacks a strong recent citation. Please add a 2023–2025 reference to support this claim.
Comment 2 (Introduction, lines 47–52):
The hypothesis that furosemide independently worsens uremic toxin clearance is stated strongly. Consider including more detailed physiological justification and at least one recent systematic review to support this.
Comment 3 (Methods, lines 258–269):
Furosemide exposure is defined only by prescription records. This may lead to misclassification bias. Please explicitly acknowledge the lack of adherence verification and discuss its implications in the limitations section.
Comment 4 (Methods, lines 270–289):
The explanation of the “symptom score 8” is rather lengthy. Consider moving the detailed calculation steps into a supplementary table and simplifying the main text.
Comment 5 (Results, lines 67–75):
The description of individual symptoms is repetitive (e.g., “washed out or drained” appears in both Results and Discussion). Consider summarizing these findings in a supplementary table or figure to improve readability.
Comment 6 (Results, lines 112–123):
The mediation effect of uremic toxins (3–5%) is modest. Please provide a more critical interpretation of whether this effect has any real clinical significance.
Comment 7 (Discussion, lines 147–157):
The statement that the association between furosemide and symptom burden “was expected” should be nuanced. Please clarify which symptoms are clearly attributable to the drug and which remain controversial.
Comment 8 (Discussion, lines 185–193):
When suggesting torsemide or bumetanide as alternatives, please add recent references (2023–2025) comparing loop diuretics in CKD, to strengthen the recommendation.
Comment 9 (Discussion, lines 217–229):
The explanation regarding OAT1/OAT3 transporter competition is highly technical. Consider simplifying and focusing more on the clinical implications for a broader readership.
Comment 10 (References):
The reference list is mostly up to date and follows MDPI guidelines, but some entries need adjustment. For example, Reference 8 should be reformatted according to MDPI style for online books: In: StatPearls [Internet]. Treasure Island (FL): StatPearls Publishing; 2024. Available from: URL. Please review all references for consistency (journal title abbreviations, sentence case in article titles, inclusion of DOIs whenever possible).
Round 2
Reviewer 1 Report
Comments and Suggestions for Authors
No further comments
Author Response
Thank you.
Reviewer 2 Report
Comments and Suggestions for Authors
Thanks for all responses! I would suggest to slightly rephrase title of the article due to a relatively weak mediating effect ("The sum of PBUTs explained 3.78% (95%CI 0.10 – 18.01%) of this association")
Author Response
Thanks! We rephrase the title as "Furosemide and the Symptom Burden: The Potential Mediating Role of Uremic Toxins in Patients with CKD"